# Cross-species transcriptomics reveals bifurcation point during the arterial-to-hemogenic transition

Shaokang Mo [1,2,3,4], Kengyuan Qu[2,3,4], Junfeng Huang [2,3,4✉], Qiwei Li[2,3], Wenqing Zhang [1✉] & Kuangyu Yen [2,3✉]

Hemogenic endothelium (HE) with hematopoietic stem cell (HSC)-forming potential emerge from specialized arterial endothelial cells (AECs) undergoing the endothelial-to-hematopoietic transition (EHT) in the aorta-gonad-mesonephros (AGM) region. Characterization of this AECs subpopulation and whether this phenomenon is conserved across species remains unclear. Here we introduce HomologySeeker, a cross-species method that leverages refined *mouse* information to explore under-studied *human* EHT. Utilizing single-cell transcriptomic ensembles of EHT, HomologySeeker reveals a parallel developmental relationship between these two species, with minimal pre-HSC signals observed in *human* cells. The pre-HE stage contains a conserved bifurcation point between the two species, where cells progress towards HE or late AECs. By harnessing *human* spatial transcriptomics, we identify ligand modules that contribute to the bifurcation choice and validate CXCL12 in promoting hemogenic choice using a *human* in vitro differentiation system. Our findings advance *human* arterial-to-hemogenic transition understanding and offer valuable insights for manipulating HSC generation using in vitro models.

[1] Division of Cell, Developmental and Integrative Biology, School of Medicine, South China University of Technology, Guangzhou, China. [2] State Key Laboratory of Experimental Hematology, National Clinical Research Center for Blood Diseases, Haihe Laboratory of Cell Ecosystem, Institute of Hematology & Blood Diseases Hospital, Chinese Academy of Medical Sciences & Peking Union Medical College, Tianjin 300020, China. [3] Tianjin Institutes of Health Science, Tianjin 301600, China. [4] These authors contributed equally: Shaokang Mo, Kengyuan Qu, Junfeng Huang. ✉email: jfhuang.dg@gmail.com; mczhangwq@scut.edu.cn; kuangyuyen@ihcams.ac.cn

Hematopoietic stem cells (HSCs) can develop into all blood cell lineages and are vital to individual survival[1,2]. The first embryonic HSC emerges in the AGM through EHT[3], whereby individual HE becomes round form HCs that aggregate into intra-aortic clusters (IACs)[4–13]. Within those HCs, CD41+CD45- T1 pre-HSCs further differentiate into CD41+CD45+ T2 pre-HSCs before maturing into definitive HSCs[14,15].

During *mouse* EHT within the AGM, not all AECs differentiate into HSC-forming HEs; some may develop into mature arterial cells[16]. Moreover, studies indicate that immature HEs need to go through an arterialization process before differentiating into definitive lymphoid–myeloid progenitors[9,17]. These findings indicate an intimate relationship between arterial specification and HSC formation[7]. Furthermore, research suggests that the transition from AEC to HE passes through a relatively unexplored intermediate pre-HE stage[18], where cells initiate hematopoietic programs while retaining arterial features. Within this AEC to pre-HE to HE transition, a developmental bottleneck exists between pre-HE and HE[18]. Runx1, a master regulator for EHT, has been shown to assist pre-HEs in overcoming this bottleneck and allowing cells to further develop into HEs[18]. These findings suggest that pre-HE may serve as a critical stage for hemogenic fate determination, and AECs require driving forces to achieve hemogenic fate. Nevertheless, the mechanisms that specifically facilitate the hemogenic choice of *mouse* AEC remain poorly understood.

*Human* HSC-forming HEs may also originate from AECs[9,17,19,20], and the existence of a *human* pre-HE stage has been suggested[20]. Nonetheless, it remains unclear whether the role of the pre-HE stage during the AEC-to-HE transition is similar between *human* and *mouse*. However, for ethical reasons, details regarding the *human* AEC-to-HE transition are less well characterized compared to that of the *mouse*. Furthermore, the *mouse* has long served as a model organism to study various *human* biological processes, including EHT[11,21,22]. This highlights the need for cross-species comparative studies that identify cellular differences and explore developmental relationships between species[23–31], thereby bypassing the limitations constraining *human* EHT research.

For cross-species comparative studies, homologous genes that share similar DNA sequences and functions across species provide an entry point. Various cross-species analysis tools have been developed utilizing homologous genes[24,25,32–34]. For instance, La Manno et al.[24] used a Bayesian generalized linear model (GLM) to identify significantly expressed genes in each cell type and compared analogous cell types across tested species using these genes. However, these approaches are restricted in cases where the annotation of corresponding cell types is unavailable. This requirement is frequently obstructed by subjective assumptions and insufficient markers for cell type identification, especially in non-model species, which constrains cross-species analysis. Consequently, a tool that requires no prior knowledge could circumvent these limitations.

To further elucidate the AEC-to-HE transition, here we introduce HomologySeeker, a cross-species analysis pipeline that detects homologous genes exhibiting highly variable expression in an unbiased manner. Without prior cell type annotation in reference or query species, HomologySeeker accurately captures well-established EHT-related homologous genes between *mouse* and *human* EHT ensembles that are constructed from publicly available single-cell transcriptome profiles. We present evidence to show that *mouse* and *human* EHT exhibit analogous cell type correspondences, with minimal T1/2 pre-HSC signals observed in *human* cells. Furthermore, *mouse* and *human* exhibit similar developmental trajectories from arterial to hematopoietic groups

and display comparable transcriptional expression patterns along the trajectories. Additionally, the pre-HE stage serves as a bifurcation point where cells face hemogenic or arterial choices, and this bifurcation point is conserved between both species. We further examine publicly available *human* spatial transcriptomics data to identify the ligand modules responsible for the distinct developmental choices of cells in the pre-HE stage between hemogenic and arterial fates. Using a *human* in vitro hematopoietic differentiation system, we validate the role of CXCL12 cytokine, identified from the module that facilitates further development into the hemogenic fate, in promoting the hemogenic choice of hemogenic precursors. Furthermore, we observed an increased production of HPCs with enhanced multilineage differentiation capability in the CXCL12 group compared to the control group. Our results contribute to a deeper understanding of *human* AECs and their selection of the hemogenic fate in vivo. Moreover, HomologySeeker provides a valuable tool for comparative transcriptomic studies across various contexts.

## Results

**HomologySeeker identifies EHT-related highly variable homologous genes**. To bypass the requirement of prior annotation, we developed an analysis pipeline called HomologySeeker ("Methods"). HomologySeeker utilizes highly variable expressed homologous genes (herein termed Homologous-HVGs), assuming that genes with high expression variability are more likely to represent genuine biological variation[35]. Briefly, HomologySeeker identifies homologous genes among tested species, ranks them based on expression variance, and then sets a cutoff using the mean value of all variances to retrieve genuine Homologous-HVGs (Fig. 1a; "Methods"). The calculation of Homologous-HVGs proceeds in an unsupervised manner, requiring no additional information, and is applicable for downstream analysis, thus offering the potential for flexible cross-species analysis application.

To evaluate the performance of HomologySeeker, we re-analyzed La Manno et al.[24] scRNA-seq datasets with pre-assigned cell identities removed (Supplementary Fig. 1a; "Methods"; Supplementary Data). We first identified overlapping Homologous-HVGs between *human* and *mouse* and then calculated the transcriptome correlation among cell clusters (Supplementary Fig. 1b, c). We then assigned identities to these cell clusters based on the expression level of the marker genes used in La Manno et al.[24]. Although we observed nearly 50% overlap between our Homologous-HVGs and the homologous genes identified by La Manno et al. (Supplementary Fig. 1d), the transcriptome correlation analysis using Homologous-HVGs reproduced the same paired cell types described in La Manno et al. (Supplementary Fig. 1e, left heatmap). This demonstrated the feasibility of Homologous-HVGs for comparative analysis across species.

We then applied HomologySeeker to identify Homologous-HVGs in *mouse* and *human* EHT ("Methods"). To encompass cells at various EHT stages before screening Homologous-HVGs, we constructed *human* and *mouse* EHT ensembles using published single-cell RNA-seq datasets generated from surface markers-enriched endothelial cells (ECs), hemogenic ECs (HECs), IACs, hematopoietic stem/progenitor cells (HSPCs), and fetal liver HSCs (FL-HSCs) ("Methods"; Supplementary Data). Directly merging datasets caused cells to cluster based on the dataset rather than cell type (Supplementary Fig. 2a). To mitigate batch effects among various datasets, we employed the "anchor"-based integration method[36] for merging datasets. Using the cell identities defined by the original studies (hereafter called pre-defined)[18–20,37,38], similar cell types tended to cluster together, prompting us to unify corresponding cell types

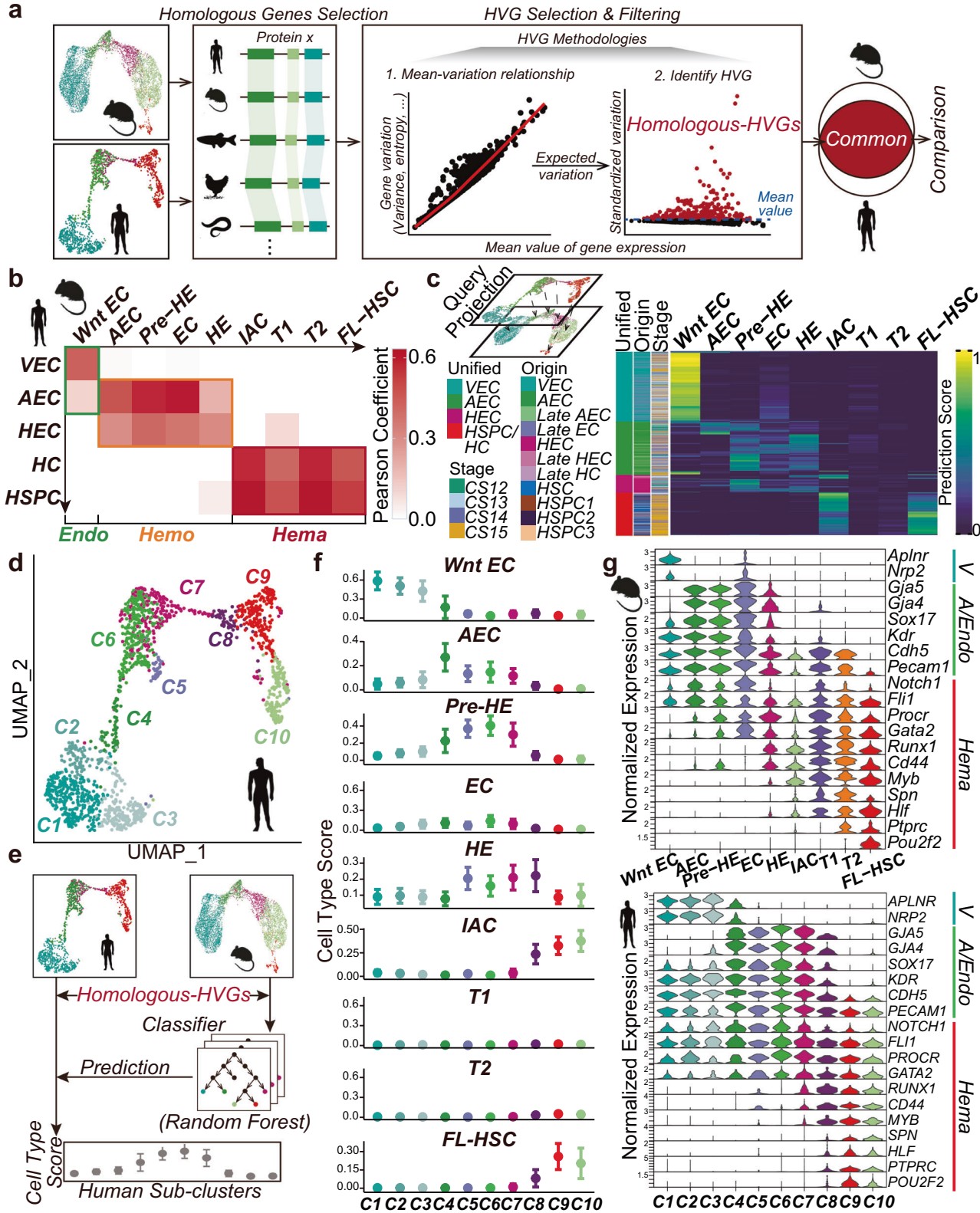

(Supplementary Fig. 2b). We observed a continuous landscape in both *mouse* and *human* EHT ensembles, as shown in the two-dimension UMAP (Supplementary Fig. 2c, d). Notably, the *mouse* EHT ensemble captured the accumulated "bulge" (Supplementary Fig. 2c), identified as the pre-HE stage in Zhu et al.[18]. These results indicate that our data merging preserved the biological relationships among cells without distorting the original datasets.

Using HomologySeeker, we identified 2456 and 3248 Homologous-HVGs in *mouse* and *human* ensembles, respectively (Supplementary Fig. 3a; Supplementary Data). As expected, EHT-associated genes, including *SOX17*[39], *RUNX1*[8,40,41], and *MYB*[42–44], appeared as Homologous-HVGs (Supplementary Fig. 3a). Among these Homologous-HVGs, 1628 genes are common between *mouse* and *human* (hereafter as "common

**Fig. 1 Cell correspondences between mouse and human EHT cells. a** HomologySeeker workflow. **b** Pearson correlation heatmap showing the relationship between *mouse* and *human* cell types, with colors representing Pearson correlation coefficients. Endo, endothelium section; Hemo, arterial/hemogenic section; Hema, hematopoietic section. **c** Top-left: Schematic of query projection. Bottom: Heatmap displaying projection scores of *human* cell types assigned new identities based on *mouse* reference. Columns represent *human* cells; rows represent *mouse* cell types; color intensity represents prediction scores. Unified: combined *human* cell type annotation; Origin: original *human* dataset annotation; Stage: cell timepoints. **d** UMAP visualization of *human* sub-clusters (C1–C10) post-unsupervised clustering. **e** Schematic of SingleCellNet model from (Tan et al.[47]). **f** SingleCellNet prediction scores of *human* cells using *mouse* cell types as a reference; higher scores indicate greater confidence. **g** Expression levels of marker genes across *mouse* cell types (upper panel) and *human* sub-clusters (lower panel). V, venous; A/Endo, arterial or endothelial; Hema, hematopoietic.

---

EHT-associated Homologous-HVGs in Supplementary Fig. 3a). In this gene set, we found that 76 out of the top 100 biological pathways enriched in *mouse* and *human* were identical (Supplementary Fig. 3c; Supplementary Data), suggesting that a large proportion of these common EHT-associated Homologous-HVGs may participate in similar biological processes. Nonetheless, species-specific terms such as "Coagulation" in *human* and "Wound healing" in *mouse* that are both related to the tissue healing process were also observed. These differences could reflect underlying biological differences or variations in nomenclature between species.

**Mouse and human EHT display analogous developmental relationship**. To evaluate the similarities between *mouse* and *human* EHT, we first examined cellular correspondence. We calculated the transcriptome correlation between the pre-defined cell types of *mouse* and *human* EHT ensembles using the common EHT-associated Homologous-HVGs ("Methods", Supplementary Fig. 3b). Based on the relative levels of Pearson correlation coefficients, we grouped corresponding *mouse* and *human* pre-defined cell types into three sections on the resulting heatmap: Endo (venous group), Hemo (arterial/hemogenic group), and Hema (Hematopoietic group) (green, orange, and red rectangles, respectively in Fig. 1b). We also conducted "anchor"-based query projection to assign potential *human* cell identities, using the *mouse* as a reference (Fig. 1c). Consistent with the transcriptome correlation analysis, almost all *human* VECs were anchored to *mouse* Wnt EC, *human* AEC/HEC to *mouse* AEC/pre-HE/HEC, and HSPC/HC to *mouse* IAC/FL-HSC (Fig. 1c).

Interestingly, using pre-defined annotation, we found that the majority of *human* HSPC1 (~69%, GJA5+ HSPC) and HSPC3 (~79%, GFI1B⁺ HSPC) exhibited higher *mouse* FL-HSC scores (Fig. 1c, Supplementary Data). The other *human* cell types within the Hema group exhibited either higher IAC or FL-HSC scores, indicating diverse hematopoietic potentials within these cells. Notably, a negligible number of *human* cells were anchored to *mouse* T1/T2 pre-HSC (T1/T2). Since the "anchor"-based projection relies on the shared nearest neighbors (SNN) of reference cells[36], we speculated whether the limited "*human* T1/T2" signals in "anchor"-based query projection resulted from the large number of *mouse* IACs co-occupying the T1/T2 in UMAP (Supplementary Fig. 4a, upper panel). However, even with a modified *mouse* EHT ensemble excluding IACs, we noted minimal T1/T2 assignment of *human* cells (Supplementary Fig. 4b, lower panel). Considering our EHT ensembles include diverse datasets with various marker combinations for isolating specific cell populations, the weak T1/2 signal observed in *human* cells may be due to the variability in marker usage and the limited presence of certain cell types (Supplementary Data). Therefore, the existence of pre-HSCs in *human* EHT remains uncertain, necessitating further exploration.

The observation that each *human* pre-defined cell type correlates with several *mouse* pre-defined cell types indicates heterogeneity within *human* cell types. To delineate potential cell sub-clusters within those heterogeneous pre-defined *human* cell

types, we re-segmented the *human* EHT ensemble using the Louvain algorithm, a graph-based unsupervised clustering method[45] ("Methods"). We employed a "cluster tree"[46] to objectively choose a stable clustering resolution, resulting in 10 sub-clusters (Supplementary Fig. 5a; Fig. 1d; C1–10). The correlation between *mouse* cell types and *human* sub-clusters reveals additional details ("Methods", Supplementary Fig. 5b). For example, *human* C4 exhibits the strongest correlation to *mouse* AEC while *human* C6 has the highest correlation to *mouse* pre-HE (Supplementary Fig. 5b). We then tried to assign cell identities to *human* sub-clusters, using the *mouse* as a reference ("Methods"). We applied a machine learning algorithm[47] to obtain cell type signatures by using the expression levels of common EHT-associated Homologous-HVGs from *mouse* to train prediction scores for *human* sub-clusters (Fig. 1e). We observed high AEC, pre-HE, and HE scores, but low T1, T2, IAC, and FL-HSC scores in *human* C4–C7 sub-clusters (Fig. 1f). Notably, *human* C6 was assigned the highest pre-HE score compared to all other *human* sub-clusters. Conversely, C8–10 exhibited low scores in AEC, pre-HE, T1, and T2 but high IAC and FL-HSC scores (Fig. 1f, Supplementary Fig. 4c). Furthermore, *mouse* pre-defined cell types and *human* re-defined sub-clusters displayed comparable EHT marker gene expression patterns (Fig. 1g).

Other than cellular correspondence, we then constructed developmental trajectories for *mouse* and *human* EHT ensembles using Monocle3[48]. Given the arterial origin of the definitive HSCs from the AGM region[49,50], we assigned *mouse* AECs and *human* C4 as the trajectory roots (Fig. 2a, b). *Mouse* and *human* EHT ensembles both displayed a continuous trajectory from AEC/C4 to FL-HSC/C10, respectively. This continuous trajectory aligns with the previous findings[18–20,37,38]. We noted that the *human* developmental trajectory diverged toward C9 and C10 (Fig. 2b). Although both C9 and C10 displayed high IAC scores, C9 had a higher FL-HSC score (Fig. 1f). To investigate the hematopoietic potential of C9 and C10, we utilized marker gene sets identified from the DEGs inference method on publically available hematopoietic progenitor cell (HPC) transcriptome profiles[51] ("Methods"; Supplementary Fig. 5b). C9 consistently displayed higher hematopoietic stem cell/multipotent progenitor (HSC/MPP) scores, as well as higher scores for LMPP1/2 that associated with monocyte/dendritic progenitors (MD) and granulocyte-monocyte progenitors (GMP) (Supplementary Fig. 5c, lower panel). Conversely, C10 showed higher lymphoid-primed MPPs (LMPP3) scores, indicating its potential for differentiation toward the lymphoid lineage (Supplementary Fig. 5c, red circle). These results suggest that C9 and C10 likely arise independently from a common precursor (C8) and possess distinct hematopoietic potentials. This finding is consistent with recent research in *mouse* that HSCs and hematopoietic progenitors may be generated independently of the heterogeneous pre-HSPC population[52,53].

As transcription factors (TFs) play a vital role in EHT[21], we further investigated the behavior of TFs along this trajectory. We first selected TFs from Homologous-HVGs (226 and 274 TFs

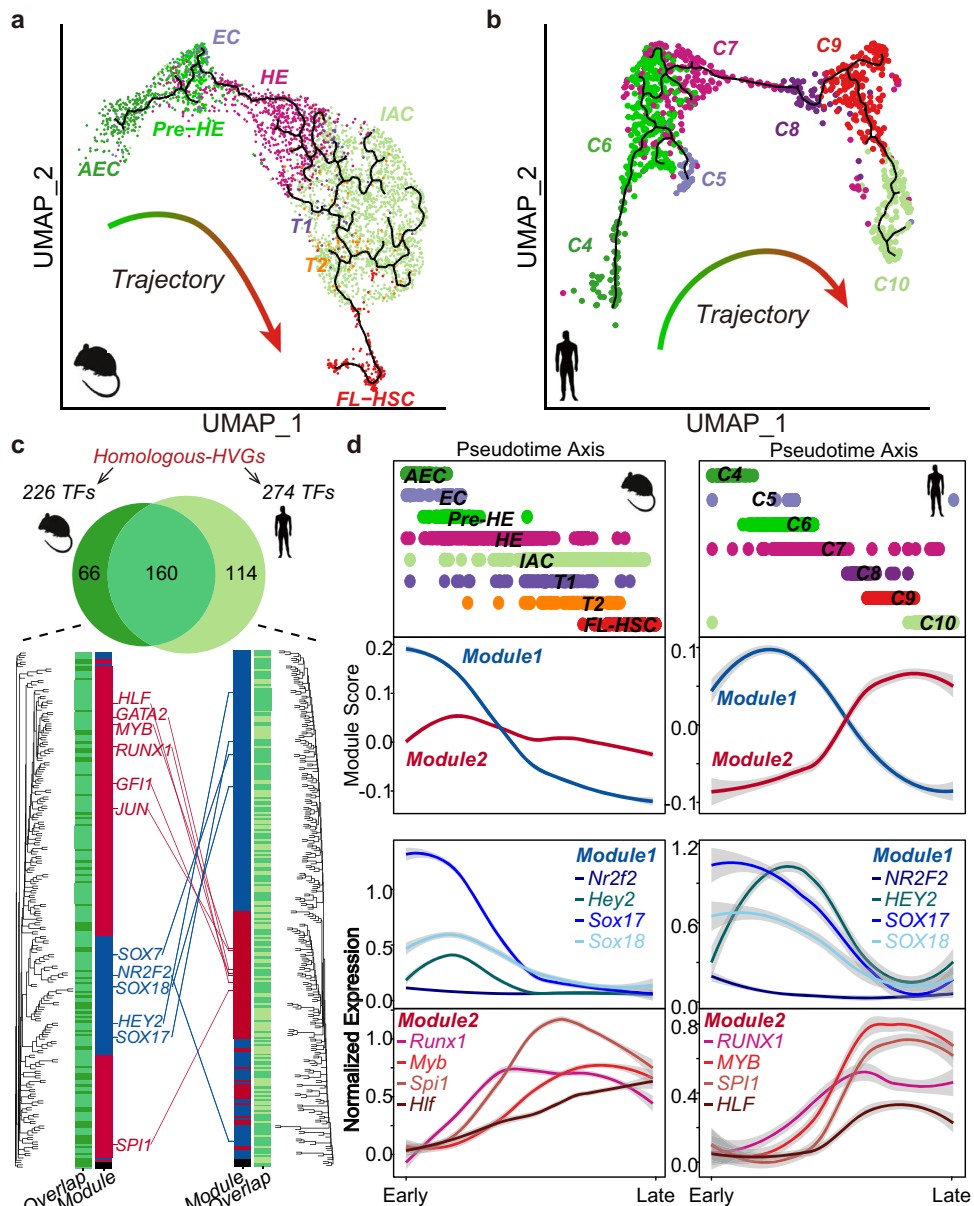

**Fig. 2 Developmental trajectories and transcriptional patterns in mouse and human EHT. a, b** Developmental trajectory of the *mouse* (**a**) and *human* (**b**) EHT ensembles from AEC/C4 to hematopoietic groups, with black lines inferring developmental paths. **c** Identification of TF modules from Homologous-HVGs set. Venn plot shows the overlap of *mouse* and *human* TFs derived from Homologous-HVGs. Lower Pannal: Hierarchical clustering of *mouse* and *human* TFs (66 + 160, 114 + 160, respectively), with the blue section representing TF module 1 and the red section representing TF module 2. **d** TF expression patterns along developmental trajectories. Upper panel: Pseudo-time changes in *mouse* and *human* developmental trajectories. Middle panel: Module scores of *mouse* and *human* TF modules along the pseudo-time axis. Lower panel: Expression patterns of representative TFs from two modules.

from *mouse* and *human*, respectively) ("Methods"; Fig. 2c). More than 50% of these TFs are common to both species. Numerous well-known EHT regulators, including endothelial/arterial TFs *SOX17/18* and hematopoietic TFs *RUNX1/MYB*, are present among these shared TFs (Fig. 2c, upper panel). The combination of both endothelial and hematopoietic TFs highlights the simultaneous regulation of endothelial and hematopoietic programs in EHT[16,19,20,37,38]. To identify potential TF regulatory modules, we applied hierarchical clustering to analyze the expression changes of these TFs along the EHT trajectory and used DynamicTreeCut[54] to discern TF modules (Fig. 2c, lower panel). We found that both species displayed two distinct TF modules. To assess the behavior of these two TF modules behave along the trajectory, we assigned module scores to each cell type

("Methods"; Fig. 2d). For both species, TF module 1 displayed a downward trend along the trajectory, indicating the down-regulation of this module as EHT progresses. This TF module 1 contains *Nr2f2*, *Hey2*, *Sox17*, and *Sox18*, with the majority being endothelial marker genes. TF module 2 comprises *Runx1*, *Myb*, *Spi1*, and *Hif*, all recognized as positive hematopoietic regulators. TF module 2 showed a consistent pattern along the *mouse* EHT trajectory while exhibiting an increasing trend in the *human* EHT trajectory. Together, these results indicate that *mouse* and *human* EHT share similarities in corresponding cell types, developmental trajectories, and transcriptional expression patterns.

**Mouse and human EHT harbor a bifurcation point during the AEC-to-HE transition.** During *mouse* EHT, subsets of AECs

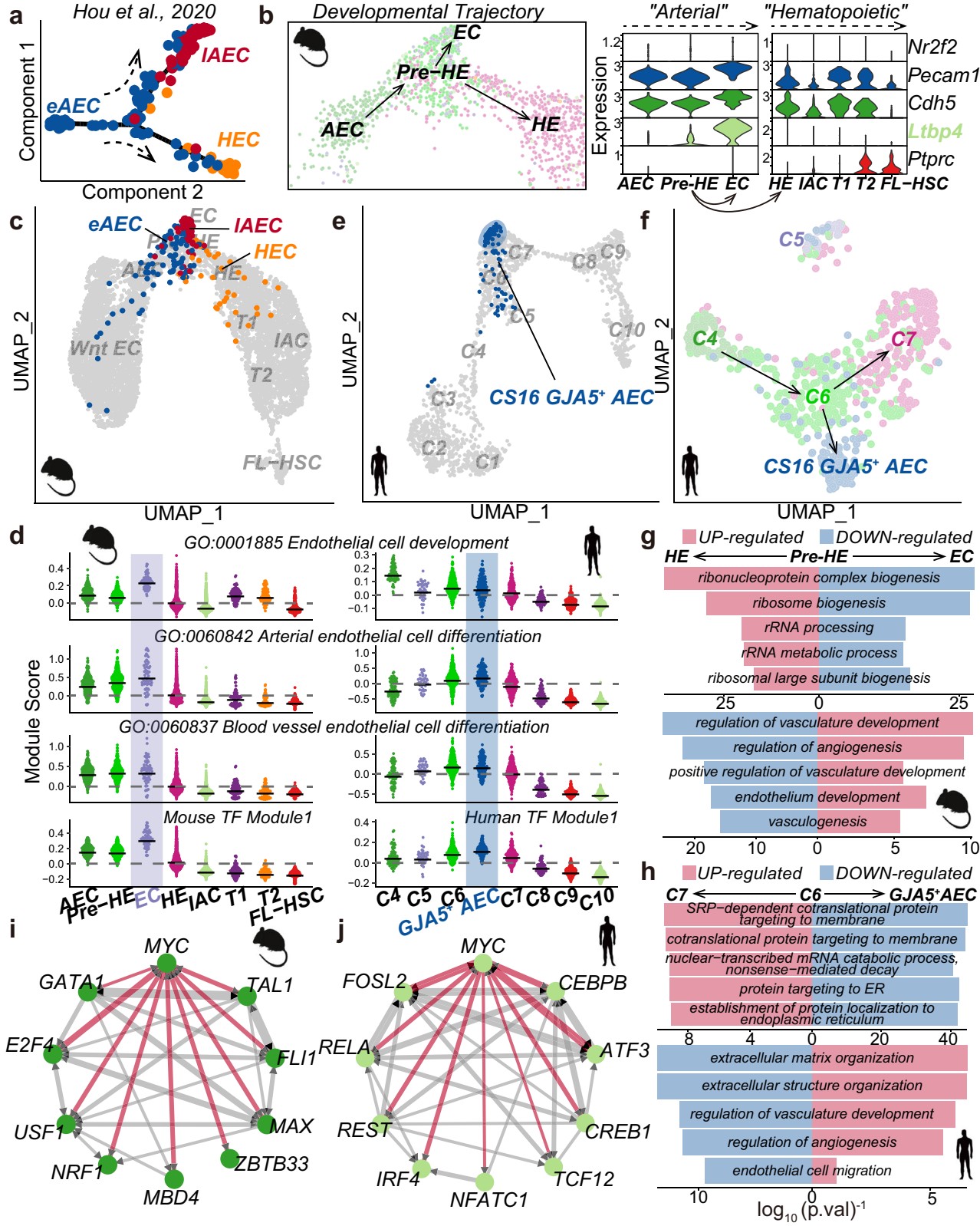

undergo cell fate choices towards either HE or mature arterial fate (late AEC, lAEC)[16] (Fig. 3a). Within the *mouse* ensemble, we observed a bifurcation in the EHT trajectory where pre-HE diverged towards HE or E11.5 EC (hereafter EC) (Fig. 2a; zoomed in Fig. 3b, left). This EC exhibited high expression of *Ltbp4*, a mature arterial feature gene[55] (Fig. 3b, right), leading us to hypothesize that pre-HE may be a bifurcation point for cell fate

decisions. To investigate this hypothesis, we projected publicly available single-cell transcriptome profiles, which functionally validate the cell fate choices of early AEC (eAEC) toward HE or lAEC[16], onto our *mouse* EHT ensemble ("Methods"; Fig. 3a). The published eAEC-to-HE trajectory aligned with our AEC-to-HE trajectory (Fig. 3c), and areas where eAEC makes choices between HE or lAEC fates corresponded to pre-HE in our data, indicating

**Fig. 3 Bifurcation point identification during the AEC-to-HE transition. a** Developmental trajectory starts from early AEC (eAEC) and bifurcates toward late AEC (lAEC) or HEC. **b** Left: Close-up of *mouse* trajectory concerning AEC-Pre-HE-EC/HE transition from Fig. 2a. Right: Expression levels of marker genes for *mouse* cell types-*Nr2f2* (venous marker), *Pecam1* (CD31-coding gene) and *Cdh5* (endothelial markers), *Ltbp4* (mature arterial marker), *Ptprc* (CD45-coding gene, hematopoietic marker). **c** Query projection of *mouse* scRNA-seq data from (Hou et al.[16]) into *mouse* ensemble (gray). eAEC (blue), early AEC; lAEC (red), late AEC; HEC (orange). **d** Module scores for *mouse* and *human* cell types/sub-clusters based on gene sets from GO terms and TF module 1. **e** Query projection of *human* GJA5+ AEC from Crosse et al.[27] into *human* ensemble (gray). **f** *Human* developmental trajectory starting from C4 and branching at C6 toward GJA5+ AEC and C7. **g** Top five biological pathways enriched by DEGs between *mouse* pre-HE and HE, and analogous terms enriched by DEGs between pre-HE and EC. Red/blue GO terms indicate enrichment by up/down-regulated DEGs. **h** Top 5 biological pathways enriched by DEGs between *human* C6 and C7, and matching terms enriched by DEGs between C6 and GJA5+ AEC. Red/blue GO terms signify enrichment by up/downregulated DEGs. **i, j** Local regulatory networks among the top ten upstream regulators of upregulated DEGs between *mouse* pre-HE and HE (**i**), and between *human* C6 and C7 (**j**). Network edges represent co-regulatory relationships with edges involving MYC, highlighted in red.

---

that pre-HE possesses a transcriptome comparable to eAEC. In addition, lAEC was projected to the tip of the "bulge" where the EC is located. EC showed the highest module scores measuring the expression level of genes involved in EC development, arterial EC differentiation, and blood vessel EC differentiation (Fig. 3d, left panel), supporting the arterial fate choice of late AEC. Moreover, we observed parallel patterns in the enriched biological pathways during the transition from pre-HE to HE/EC and from early AEC to HE/late AEC (Supplementary Fig. 8a, b). These results indicate that the eAEC identified in previous work[16] may correspond to the pre-HE in our *mouse* EHT ensemble; consequently, pre-HE in our *mouse* EHT ensemble may possess the ability to choose different cell fates.

We also observed a bulge between *human* C6 and C7 sub-clusters (Fig. 1d). Given the similarity between these two species (Figs. 1 and 2), *human* EHT might encounter similar cell fate choices during the AEC-to-HE transition. Notably, *human* C6 showed the highest pre-HE score among all other sub-clusters (Fig. 1f) and exhibited pre-HE signatures in terms of dynamic trajectory, marker genes, and cell composition[20] (Supplementary Fig. 6a–c). These findings indicated that C6 likely represents *human* pre-HE and faces bifurcation choice akin to *mouse* pre-HE. To explore this, we projected published GJA5+ AECs from CS16 dorsal aorta[27] (hereafter called GJA5+ AEC) onto the *human* EHT ensemble using a similar approach to Fig. 2b ("Methods"). Most GJA5+ AECs clustered at the tip of the C6/C7 bulge, with some scattered at C5/C6 (Fig. 3e), resembling *mouse* lAECs (Fig. 3c). We then assessed whether C6 exhibits diverging trajectories toward C7 or GJA5+ AEC. After merging *human* C4-C7 and GJA5+ AEC into a localized cohort, we performed Monocle3 trajectory analysis using a similar strategy to Fig. 2a, b ("Methods"). Comparable to the bifurcation choices encountered by *mouse* pre-HE, we observed a trajectory starting from C4, reaching C6, and then diverging into C7 or GJA5+ AEC (Fig. 3f; Supplementary Fig. 7a, b). The transition from C6 to C7 revealed the emergence of hemogenic markers (*RUNX1*[8,40,41] and *KCNK17*[20]) (Supplementary Fig. 7d), whereas the C6 to GJA5+ AEC transition retained pre-HE markers[20] *ALDH1A1* and *IL33* but lacks expression of hematopoiesis-associated genes (*HOXA9*[56] and *MLLT3*[57]). This indicates that the transition from C6 to C7 is involved EHT, while the C6 to GJA5+ AEC transition follows arterial processes. Similar to *mouse* ECs, *human* GJA5+ AECs exhibited comparatively high scores in all tested modules (Fig. 3d, right panel). Notably, *human* C5 appeared as an outlier, suggesting that C5 might not participate in EHT either (Fig. 3f).

We further examined if similar transcriptional networks govern the fate choices in *mouse* pre-HE and *human* C6 sub-cluster. We integrated GJA5+ AECs into our *human* EHT ensemble (Supplementary Fig. 7c) and calculated the differentially expressed genes (DEGs) between these two choices (Supplementary Fig. 8c, d; Supplementary Data, pre-HE vs. HE/EC in *mouse*,

C6 vs. V7/GJA5+ AEC), followed by GO term analysis (Fig. 3g, h; Supplementary Data). Upregulated DEGs in *mouse* pre-HE-to-EC and *human* C6-to-GJA5+ AEC transition were enriched for vasculature/angiogenesis development and endothelial development pathways, indicating a vascular fate toward EC/GJA5+ AEC (Fig. 3g, h). These pathways were down-regulated in the pre-HE-to-HE transition in *mice* and the C6-to-C7 transition in *humans* (Fig. 3g, h), suggesting an alternative choice toward HE/C7 direction. Upregulated DEGs of the pre-HE-to-HE transition in *mouse* were mainly enriched for ribosome-related pathways, while in *human*, they predominantly focused on protein modification-related pathways. (Fig. 3g, h, bar plot in red). This is consistent with prior research that highlighted the role of enhanced ribosomal activity and protein translational processes in the development of HSC-primed HE across both species[16,19]. We then employed ChEA3[58] analysis to identify potential upstream regulators of these upregulated DEGs in both the pre-HE-to-HE in *mouse* and the C6-to-C7 in *human*. We observed that the top 10 regulators converge on the core factor MYC[59] in both species (Fig. 3i, j; Supplementary Data), aligning with a prior study that demonstrated diminished HECs in the aorta upon *Myc* deletion[58]. These results indicated that parallel transcriptional networks govern the fate transitions in both *mouse* and *human*.

**Identification of external signals that facilitate bifurcation choices during the AEC-to-HE transition.** Cell fate transition during EHT is guided by the surrounding cellular environment[60]. To gain a better understanding of how external factors impact transcription networks during bifurcation choice, we analyzed publicly available *human* spatial transcriptomics data, which provided transcription profiles for the nearby niche of AGM[20]. We treated each spot on the spatial transcriptomics slide 7 from the CS15 *human* embryo as a single pseudo cell and used applied unsupervised clustering[45] to categorize these pseudo cells into 11 major cell populations, with S1 and S8 derived from AGM (Supplementary Fig. 9a; "Methods"). We then applied NicheNet[61], which predicts ligand-target connections through an integrated model encompassing the signaling path from ligands to target genes, to identify potential ligands using DEGs from C6-to-C7 and C6-to-CS16 GJA5+ AEC as downstream targets ("Methods"; Fig. 4a; Supplementary Data). We considered potential ligands for EHT as true only if they were expressed by the S1 or S8 cell populations (Supplementary Fig. 9b, c; Supplementary Data). Notably, various ligands that facilitate C6 to choose distinct fates could affect the same downstream targets (Supplementary Fig. 9c, black rectangle).

Among the true potential ligands that facilitate the C6-to-CS16 GJA5+ AEC transition, TGFB1 is the top candidate (Supplementary Fig. 9b, arterial module). However, TGFB1 also contributes to the C6-to-C7 transition, implying its divergent roles in cell fate selection, as previous work has shown that the interplay between TGFβ and Notch signaling directs AECs to adopt a hemogenic

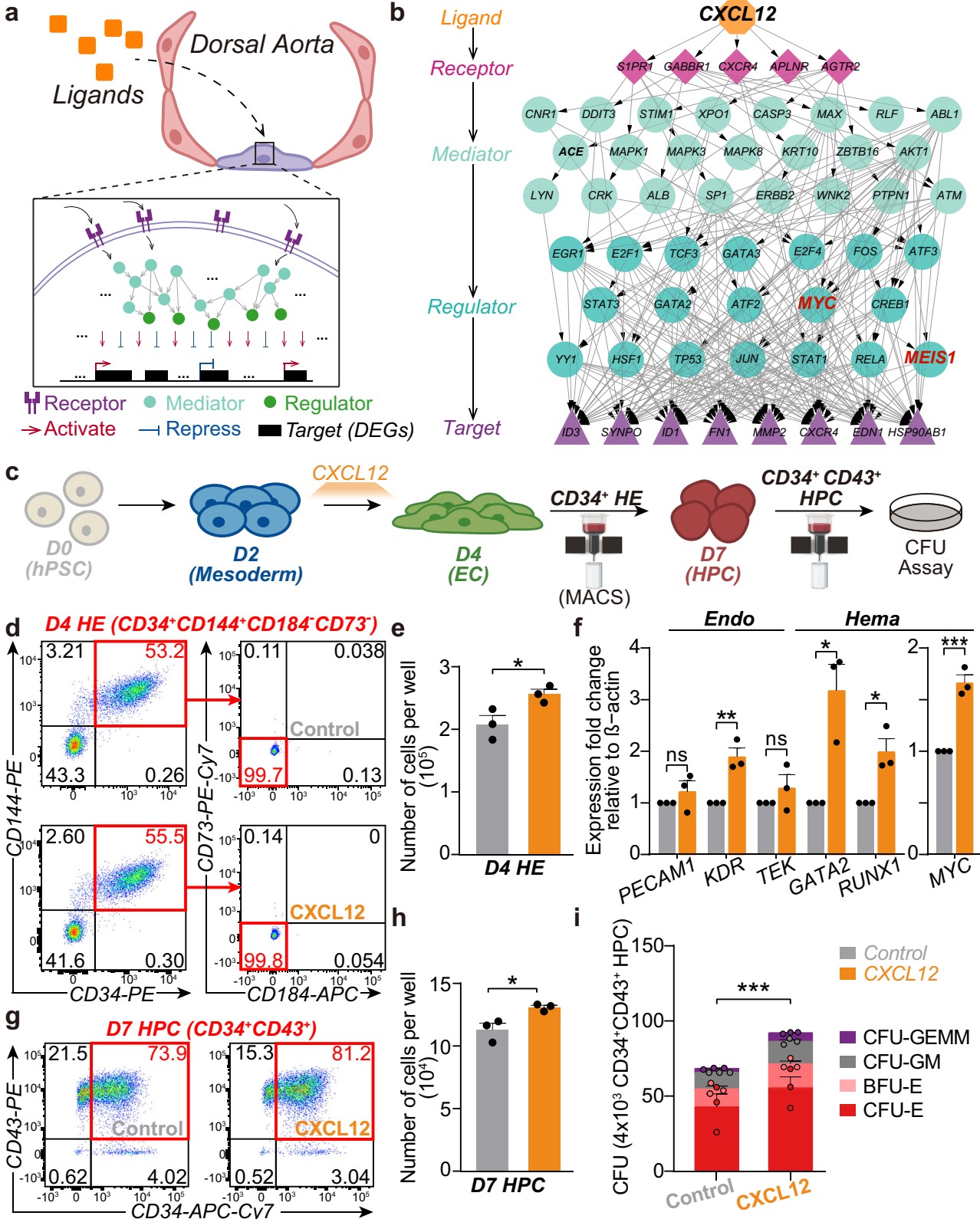

identity[62]. SPP1 has been shown to target CD44[19,27], a receptor that marks the HSPC-forming AECs[63]. Our analysis further suggests that SPP1 promotes the arterial fate of those HSPC-forming arterial ECs.

All of the top five true potential ligands that facilitate the C6-to-C7 transition (Supplementary Fig. 9b, Hemogenic module) are pivotal during EHT[62,64,65]. BMP signals (BMP4, BMP5, and

BMP7), especially BMP4, are required for HSC emergence and maturation within AGM[66–68]. VEGFA is required for NOTCH signaling, which activates the hematopoietic program[65,69–71]. Besides maintaining the quiescent HSC pool[72,73], the CXCL12-CXCR4 axis has been found to either suppress the EC program of *mouse* HE[74] or facilitate the generation of engrafting HSCs from E9-to-E10 hemogenic precursors[75], highlighting a vital role for

**Fig. 4 Identification of spatial ligands that facilitate cell fate choices of human pre-HEs. a** Schematic of signaling pathways from spatial ligands to target genes (DEGs). **b** Signaling pathway mediated by CXCL12. The signal travels from CXCL12 (ligand, orange) to receptors (pink) through signaling mediators (light blue) and regulators (blue), ending at target genes (purple) (DEGs between C6 and C7 regulated by CXCL12). **c** Schematic of *human* in vitro hematopoietic differentiation system from (Shen et al.[79]). **d** Representative flow cytometric analysis of the in vitro-defined HEs (CD34$^+$CD144$^+$CD184$^-$CD73$^-$) from day 4 differentiation. **e** Cell number of in vitro-defined HEs (CD34$^+$CD144$^+$CD184$^-$CD73$^-$) sorted from day 4 (left panel) differentiation with or without CXCL12 treatment ($n = 3$, $*P < 0.05$). **f** Expression of *PECAM1, CDH5, TEK, GATA2,* and *RUNX1* in cells from day 4 with or without CXCL12 treatment. The expression level was normalized to that of β-actin. ns, not significant. ($n = 3$, $*P < 0.05$; $**P < 0.01$). **g** Representative flow cytometric analysis of the in vitro-defined HPCs (CD34$^+$CD43$^+$) from day 7 differentiation. **h** Cell number of in vitro-defined HPCs (CD34$^+$CD43$^+$) sorted from day 7 differentiation with or without CXCL12 treatment ($n = 3$, $*P < 0.05$). **i** Colony-forming unit (CFU) assay of HPCs generated with or without CXCL12 treatment. CFUs per $4 \times 10^3$ cells plated ($n = 3$, $***P < 0.001$).

CXCL12 in EHT. PCDH7, one of the potential ligands that facilitate the C6-to-CS16 GJA5$^+$ AEC transition (Supplementary Fig. 9b), also interacts with CXCR4 (Supplementary Fig. 9c), implying a distinct function for CXCR4 in cell fate choice under different conditions.

In our ligand-target analysis, CXCL12 activates several key regulators (Fig. 4b). Among these key regulators, MEIS1 has been shown to promote the hemogenic specification of APLNR$^+$ mesoderm progenitors in *human*[76]. Consistent with this, MEIS1 was also identified as a TF that specifically regulates the DEGs between *human* C6 and C7 (Supplementary Fig. 9c, C7 vs. C6 specific). Moreover, GATA2, which is vital in HSC generation[77,78], participated in the CXCL12 signaling pathway. One of the CXCL12 target regulators is MYC, which we identified as the core upstream regulator of upregulated DEGs in both species (Fig. 3h, i). Considering its prominent role among the true potential ligands facilitating the C6-to-C7 transition, we hypothesize that CXCL12 may selectively promote the hemogenic choice in *human* embryonic hematopoietic development.

**CXCL12 promotes hemogenic fate.** Given the potential role of CXCL12 during hemogenic fate determination of pre-HE, we wondered if CXCL12 treatment can truly promote HE formation from the hemogenic precursor. To this end, we took advantage of a *human* pluripotent stem cell (hPSC) in vitro system[79] (Fig. 4c) that mimics hematopoietic differentiation, allowing us to bypass the ethical restrictions on *human* embryos. In this monolayer-based in vitro system that underwent chemically defined culture, hPSCs (Day 0, referred to as D0) progress through mesoderm (D2) and endothelial (D4) specification before developing into hematopoietic progenitors (HPCs) (D7) with multilineage differentiation capability. CD144$^+$CD34$^+$CD73$^-$CD184$^-$ cells at D4 are considered as HEs[80] (hereafter called in vitro-defined HEs), and CD34$^+$CD43$^+$ cells at D7 as HPCs (hereafter called in vitro-defined HPCs). As in vitro-defined HEs are mainly enriched at D4, hence we anticipate that the hemogenic precursor-to-HE transition happens between D2 and D4 in this system. Therefore, we added CXCL12 (Peprotech, 300-28A) into the culture medium on day 2 to determine whether it promoted the hemogenic precursor-to-HE transition. After two days of differentiation until D4, we quantified the abundance of in vitro-defined HEs (CD144$^+$CD34$^+$CD73$^-$CD184$^-$) using flow cytometric analysis (FACS) ("Methods"; Fig. 4d; Supplementary Fig. 10a). We observed that CXCL12 treatment (referred to as the CXCL12 group) significantly increased the amount of in vitro-defined HEs as compared to the control group (no CXCL12 treatment) (Fig. 4e, left panel; "Methods", $*P$ value $< 0.05$), supporting the promoting role of CXCL12 in hemogenic fate determination.

Quantitative real-time polymerase chain reaction (qRT-PCR) analyses of the D4 cells from the CXCL12 group showed significantly increased RNA expression of hematopoietic markers, such as *GATA2* and *RUNX1* (Fig. 4f; "Methods", $*P$ value $< 0.05$). Meanwhile, the RNA expression of endothelial markers, such as

*PECAM1* (CD31-coding gene) and *TEK* (TIE2-coding gene), showed no significant change (Fig. 4f). Notably, *KDR*, which encodes a VEGFR2 receptor that marked the endothelial subset with hematopoietic potential[81,82], exhibits significantly increased RNA expression under CXCL12 treatment. Additionally, the increased *MYC* expression agrees with our analysis that CXCL12 might positively regulate *MYC* during the C6-to-C7 transition (Fig. 3h, i; Fig. 4f). These results indicated that CXCL12 facilitates hemogenic fate by promoting the hematopoietic program instead of repressing the endothelial program. We then evaluated the hematopoietic potential for these HEs. Given that in both our FACS results (Fig. 4d) and the original paper[80,83], at D4, CD34$^+$ cells encompassed all CD34$^+$CD144$^+$CD73$^-$CD184$^-$ HE cells. Therefore, for subsequent hematopoietic potential analysis, we used Magnetic-Activated Cell Sorting (MACS) to isolate CD34$^+$ cells at D4 as a representation of the CD34$^+$CD144$^+$CD73$^-$CD184$^-$ HEs. These CD34$^+$ cells were cultured in STEMdiff APEL 2 medium until day 7 (D7) and followed by assessing the formation of HPCs. Using FACS, we observed a significantly higher amount of in vitro-defined HPCs (CD34$^+$CD43$^+$) from the CXC12 group (Fig. 4g, h; Supplementary Fig. 10b; "Methods", $*P$ value $< 0.05$), supporting the enhanced hematopoietic potential of those in vitro-defined HEs from the CXCL12 group.

Given the increased production of in vitro-defined HEs and HPCs, we hypothesized that CXCL12 treatment could promote the multilineage potential of those in vitro-defined HPCs. To test this hypothesis, we used MACS to sort an equal number of in vitro-defined HPCs (CD34$^+$CD43$^+$) at D7 from both the CXCL12 and control groups and followed by measuring their colony-forming potential using a colony-forming unit (CFU) assay ("Methods"). In line with our hypothesis, we observed significantly higher numbers of hematopoietic colonies of myeloid and erythroid lineages from the CXCL12 treatment group (Fig. 4i; "Methods", $***P$ value $< 0.001$), indicating an enhanced multilineage differentiation capability of HPCs from the CXCL12 group. In summary, our results not only support the role of CXCL12 in facilitating the hemogenic fate of HE precursors but also highlight its role in promoting the hematopoietic potential of HEs (Fig. 5).

**Discussion**

Here we introduce a cross-species analysis method (HomologySeeker) based on homologous genes exhibiting high levels of expression variability (Fig. 1a). Compared to state-of-the-art methods that require prior cell type annotation, including a recent model (CAME)[34] that advances the utilization of non-one-to-one homologous gene mapping, HomologySeeker avoids prior cell type annotation for cross-species comparisons. We utilize HomologySeeker to study EHT transcriptome ensembles that we constructed from publicly available single-cell RNA-seq datasets (Supplementary Fig. 2). These ensembles could serve as an expandable repository for the scientific community. We showed

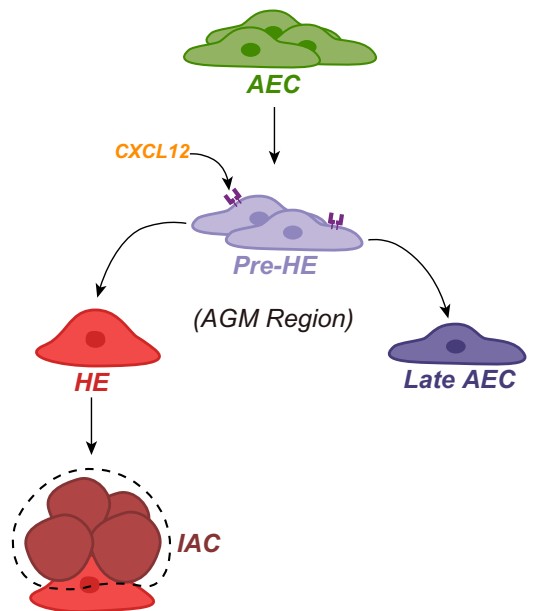

**Fig. 5 Cell fate choices during the pre-HE stage.** During *mouse* and *human* EHT, a specific subset of AECs differentiates into pre-HEs. In response to interactions between specific cell fate determinants (like CXCL12) and their corresponding receptors, pre-HEs may advance toward either a hemogenic or arterial fate. Those hemogenic precursors that acquire the CXCL12 signal can further develop into a hematopoietic population with multilineage differentiation capability aggregated in clusters. AEC arterial endothelial cell, pre-HE pre-hemogenic endothelium, HE hemogenic endothelium, IAC intra-aortic cluster.

that *human* and *mouse* EHT display analogous cell type correspondences, similar developmental trajectories, and comparable transcription expression patterns (Figs. 1 and 2), substantiating the conserved nature between these two species. However, due to the diversity of datasets and potential omissions in the data integrated into our *human* ensemble, the minimal T1/2 signals we observed in *human* cells (Fig. 1c–f; Supplementary Fig. 4) could suggest a scarce presence of pre-HSC cells if any. Moreover, we do not observe *mouse* cells equivalent to the *human* C5 population (Fig. 1f, g), warranting further exploration. We observe that pre-HEs have the potential to differentiate into either HEs or lAECs, a phenomenon conserved between *mouse* and *human* (Fig. 3), which refines our understanding of the *human* arterial-to-hemogenic transition. We further identify ligand modules that contribute to pre-HE choices and demonstrate that CXCL12 significantly enhances the HSPC-forming potential of HE precursors using an in vitro differentiation system (Fig. 4).

Our finding that *mouse* pre-HEs can further differentiate into HEs or E11.5 ECs (Fig. 3) harmonizes with parallel studies[16,18]. These studies show that the cell fate choice of eAECs toward either HEs or lAECs[16] co-occurs with the pre-HE stage, an intermediate stage between AEC and HE[18]. Zhu et al.[18] knocked down Runx1, and we observed increased pre-HE numbers but a reduced amount of HE. This pre-HE-to-HE transition is similar to that of the eAEC-to-HE transition, as reported by Hou et al.[16]. Additionally, the transcriptome profile of lAECs in Hou et al.[16] overlaps with E11.5 ECs that reside at the end of the trajectory, away from the path toward HEs (Fig. 3c). This pre-HE-to-E11.5 EC transition resembles that of the eAEC-to-lAEC transition in Hou et al.[16]. Furthermore, cells from this *human* C6 sub-cluster exhibit similar fate choices as *mouse* pre-HE towards the C7 sub-cluster, which contains HE signatures, or GJA5[+] AECs (Fig. 3f).

Taken together, the pre-HE stage may serve as a bifurcation point for cell fate decisions, and this phenomenon is conserved across species.

Our study shows that CXCL12 not only facilitates hemogenic fate but also promotes hematopoietic potential, as evidenced by the increased number of in vitro-defined HPCs, indicating the formation of genuine HSPC-forming HEs under CXCL12 treatment (Fig. 4). Furthermore, hematopoietic progenitors treated with CXCL12 exhibited enhanced multilineage differentiation capability (Fig. 4i), consistent with a previous study that CXCL12-CXCR4 signaling enables the generation of long-term engrafting HSCs from *mouse* E9-to-E10 AGM derived hemogenic precursors[75]. Additionally, our study identifies CXCL12 as a shared upstream effector for hemogenic regulators, like MEIS1[75] and MYC[59] (Fig. 4b), both involved in *mouse* hemogenic fate choice decisions. These findings suggest that CXCL12 plays a conserved role during the pre-HE-to-HE transition between *mouse* and *human*, potentially serving as a critical checkpoint for hematopoiesis manipulation. Nevertheless, further exploration is needed to elucidate the precise mechanism of how CXCL12 promotes hemogenic fate. Studies[72,73] have shown that CXCL12 functions through the CXCR4 receptor, which is also involved in EHT[9,19,74,75]. Interestingly, only a subset of the CXCR4[+] population shows hemogenic potential, while others exhibit arterial features[80]. Our ligand network analysis indicates that CXCR4 interacts only with either CXCL12 or PCDH7 (Supplementary Fig. 9c), the latter belongs to ligand modules that facilitate arterial choices in pre-HEs (Supplementary Fig. 9b, c). Consequently, PCDH7 may compete with CXCL12 for CXCR4, resulting in a mutually antagonistic relationship between CXCL12 and PCDH7 that co-regulates pre-HE fate selection.

Our work presents HomologySeeker as a new approach for investigating cell fate transitions across species. By applying this method to the study of EHT, we have advanced our understanding of this critical developmental stage, particularly during the *human* arterial to hemogenic transition. Our findings provide valuable insight into the regulation of hematopoiesis and the enhancement of hematopoietic efficiency in *human* in vitro differentiation.

## Methods

**Maintenance and hematopoietic differentiation of hPSCs.** The H1 hPSC line was obtained from the WiCell Research Institute (Madison, WI, http://www.wicell.org). Cultured cells were maintained on Matrigel-coated 6-well plates (Corning) containing E8 medium (Gibco)[84,85], and the medium was replaced daily. The hPSCs were sub-cultured every 3–4 days with a treatment of 0.5 mM ethylene-diaminetetraacetic acid (EDTA; Life Technologies) for passaging when cells reached 60–70% confluence. For hematopoietic differentiation, single hPSCs were obtained for sequential EC–HC induction. Briefly, single-cell suspensions of hPSCs were obtained by treating the hPSC cultures at 70–80% confluency with TrypLE (Thermo Fisher Scientific). Single cells were then plated at an optimized density of $6 \times 10^3$ cells/well onto 12-well plates (Corning) coated with vitronectin (Peprotech) in STEMdiff APEL 2 Medium (STEMCELL Technologies) supplemented with 3 μM GSK3 inhibitor, CHIR99021 (ABM Inc), 4 ng/ml ActivinA (Peprotech), 10 ng/ml BMP4 (Peprotech), and 10 μM Rho kinase inhibitor, Y-27632 (STEMCELL Technologies) on day 0. After 48 h (day 2), the medium was changed to STEMdiff APEL 2 Medium supplemented with 40 ng/ml VEGF (Peprotech). For the following 24 h (day 3), recombinant *Human* FGF2 (ABM Inc.) was added to a final concentration of 40 ng/ml until day 4. CD34[+] cells were isolated from differentiated cells on day 4 by magnetic-activated cell sorting (MACS, Miltenyi Biotec.). We re-seeded the isolated CD34[+] cells on vitronectin (Peprotech)-coated 12-well plates (Corning) at a density of $1.25 \times 10^5$ cells/well in STEMdiff APEL 2 Medium (STEMCELL Technologies) supplemented with 40 ng/ml VEGF (PeproTech) and 40 ng/ml FGF2 (ABM Inc) until day 7. The entire differentiation process was incubated at 37 °C in 5% $CO_2$ with 100% humidity.

**Flow cytometry analysis.** Cells were dissociated to form a single-cell suspension by TrypLE treatment and washed with FACS buffer PBE (2% FBS and 0.5 mM EDTA in PBS). The dissociated cells were then resuspended in PBE and labeled with fluorochrome-conjugated anti-*human* CD73-PE-Cy7 (BioLegend, clone: AD2), CD184-APC (Invitrogen, clone: 12G5), CD144-PE (Invitrogen, clone: 16B1),

CD34-APC-Cy7 (BioLegend, clone: 561), CD34-PE (Invitrogen, clone: 4H11), and CD43-PE (BioLegend, clone: 10G7). Dead cells were excluded according to DAPI (BD Biosciences) staining. Isotype-matched control antibodies were used to determine the background. Flow cytometry was performed using Canto II analyzer (BD Biosciences). Data analysis was performed using FlowJo software (Tree Star, Inc.).

**RNA extraction and quantitative real-time polymerase chain reaction (qRT-PCR) assay.** Total RNA was extracted using a TRizol reagent (Roche). cDNA was synthesized from 2 μg of total RNA using the GoScript™ Reverse Transcriptase Kit (Promega) and stored at -80℃ until use. Real-time PCR was performed using a ChamQ SYBR Color qPCR Master Mix (Low ROX Premixed) (Vazyme) on a QuantStudio™ 3 (Applied Biosystems). Amplification of β-actin was conducted in parallel to control for the quantity of loaded cDNA in each reaction. Primer sequences are listed in Supplementary Data.

**Hematopoietic colony-forming unit (CFU) assays.** 4000 CD34⁺CD43⁺ HPC single cells in 0.1 ml IMDM (Life Technologies) with 2% FBS were mixed with MethoCult H4034 Optimum (STEMCELL Technologies). The mixture was then transferred to ultra-low attachment 12-well plates (Corning). The cells were incubated at 37 °C in 5% $CO_2$ with 100% humidity for 14 days before counting colonies. Each type of colony was classified according to morphology. Each assay was performed in triplicate.

**scRNA-seq data collection and pre-processing.** For *mouse* and *human* midbrain scRNA-seq datasets from La Manno et al.[24], gene expression matrices deposited in the NCBI Gene Expression Omnibus (GEO) were downloaded under the accession number GSE76381 (STRT-seq).

For *mouse* EHT scRNA-seq datasets, gene expression matrices under accession numbers: GSE112642 (Baron et al.[38], Cel-seq) and GSE137117 (Zhu et al.[18], 10× Genomics droplet-based scRNA-seq) were downloaded (Supplementary Data). For the scRNA-seq data of Zhou et al.[37] (GSE135202, STRT-seq) and Hou et al.[16] (GSE139389, STRT-seq), raw reads were split by barcode sequence attached in Read 2. The TSO sequence and adapter contaminants were trimmed using trim_galore (v0.6.7)[86] for Read 1. Trimmed Reads 1 were aligned against mm10 *mouse* genome using STAR (v2.6.0c)[87] (Parameters: outFilterMatchNminOverLread = 0.3, outFilterScoreMinOverLread = 0.3). Uniquely mapped reads were counted using HTSeq (v0.13.5)[88] and grouped by the cell-specific barcodes. For each barcode, the copy number of transcripts of a given gene was taken as the number of distinct UMIs of that gene.

*Human* EHT scRNA-seq data were collected under accession numbers: GSE135202 (Zeng et al.[19], STRT-seq and 10X Genomics droplet-based scRNA-seq), GSE162950 (Calvanese et al.[20], 10× Genomics droplet-based scRNA-seq), and GSE151877 (Crosse et al.[27], 10× Genomics droplet-based scRNA-seq) (Supplementary Data). Briefly, sequencing data from 10× genomics was processed using CellRanger (v2.1.1) with default mapping arguments. The sequencing data of STRT-seq were processed as mouse STRT-seq datasets, but using the GRCh38/hg38 *human* genome for reads mapping. To keep the consistency of gene annotation, all gene names from *mouse* and *human* datasets were converted to official gene symbols using the alias2Symbol function from limma (v3.18.10)[89]. Only CDH5⁺GJA5⁺HEY2⁺APLNR⁻NR2F2⁻PDGFRA⁻PDGFRB⁻GYPA⁻EPCAM⁻ cells from the single-cell dataset (Crosse et al.) were selected as GJA5⁺ AECs.

For the 10× Genomics droplet-based scRNA-seq dataset from Huo et al.[51], the gene expression matrix was downloaded from GEO under the accession number GSE224714. Only cells sampled from healthy controls were retained for further analysis.

**HomologySeeker method.** As comparative analyses using all homologous genes may include genes that are not expressed across all cells or unrelated to the development system in question, making downstream interpretation challenging. To avoid this, we sought to take advantage of the concept of highly variable genes (HVGs), which is widely used in single-cell RNA-seq analysis to select genuine biological variations. Furthermore, HVGs can be identified in an unsupervised and low-calculation-cost manner that applies to various kinds of development systems. HomologySeeker is designed to identify homologous gene sets with highly variable expression (Homologous-HVGs) for cross-species analysis while keeping species-specific homologous/non-homologous genes for additional purposes.

HomologySeeker consists of two main steps: (i) Homologous gene collection and filtering, (ii) homologous-HVGs identification. Briefly, homologous genes between species are collected from Ensemble databases using "getLDS" function in biomaRt (v2.46.3 was used in this study). HomologySeeker only keeps genes with one-to-one orthology and high orthology confidence introduced by the Ensemble database in "Ortholog_qc_manual" section (https://ensembl.org/info/genome/compara/Ortholog_qc_manual.html). Next, returned gene sets are fed into an HVG selection method (Seurat v4.1.1[36] was used in this study) to get variation levels (i.e., standardized variance) of all genes for each species. Finally, to objectively select variable genes, HomologySeeker utilizes the mean values of the variation levels of gene sets as the cutoff for selecting "genuine" highly variable

genes, which results in species Homologous-HVG sets for further comparative analysis.

**EHT ensembles construction.** Expression matrices from Zhou et al., Baron et al., and Zhu et al. were used to construct the *mouse* EHT ensemble (Supplementary Data). Only cells annotated as venous/arterial EC, EC, HE, IAC, T1/2 pre-HSC, and FL-HSC were included. The ensemble was constructed based on the instruction of "Performing integration on datasets normalized with SCTransform"[90] (https://satijalab.org/seurat/articles/integration_introduction.html) in Seurat. Briefly, normalization and highly variable genes selection were performed for each dataset using "SCTransform" function (Parameter: method = "glmGamPoi", min_cells = 1). Integration features and objects were prepared using "SelectIntegrationFeatures" and "PrepSCTIntegration" with default settings, respectively. Then anchors identified using the "FindIntegrationAnchors" function among datasets were used for data integration using "IntegrateData" function with default parameters. The resulting integrated dataset was called the "EHT ensemble".

For *human*, expression matrices from Zeng et al. and Calvanese et al. were used to construct the *human* EHT ensemble. Only cells annotated as venous/arterial EC, HE, and HSPC/HC were included for further analysis. The *human* EHT ensemble construction was performed as a mouse EHT ensemble.

To maintain the consistency of cell annotation, *mouse* cell types are unified as Wnt_EC, AEC, EC, Pre-HE, HE, IAC, T1, T2, and FL-HSC according to original annotations, whereas *human* cell types were unified as VEC, AEC, HEC, HC, and HSPC (Supplementary Fig. 2c).

For merging GJA5⁺ AECs into the *human* EHT ensemble, STACAS (v2.0.1)[91], a sub-type anchoring correction method for alignment in Seurat, was used to prevent batch effect overcorrection. Briefly, each dataset (Zeng et al. (STRT-seq + 10×), Calvanese et al., and GJA5⁺ AEC) was normalized using the "NormalizeData" function in Seurat. Then "Run.STACAS" function (Parameters: dims = 1:50) was used to perform the integration analysis of all normalized datasets.

**Dimension reduction and unsupervised clustering.** Dimension reduction and unsupervised clustering were done by Seurat unless otherwise mentioned.

To visualize single cells in 2D space, the dimension of both EHT ensembles was first reduced based on principal component analysis using the "RunPCA" function with default settings. EHT ensembles were visualized by projecting cells in 2D space using UMAP implemented in "RunUMAP" function (Parameters: dims=1:50).

To cluster the *human* single cells, the nearest-neighbor graph of the *human* EHT ensemble was first constructed using "FindNeighbors" function, and sub-clusters were identified by the Louvain algorithm using "FindClusters" function (Parameters: resolution = 0.8, clustree (v0.4.4)[46] were used to determine the optimal clustering resolution).

For the 10x Genomics droplet-based scRNA-seq dataset from Huo et al., datasets from each healthy donor were integrated based on the instruction of "Performing integration on datasets normalized with SCTransform". Then the dimension of the integrated dataset was reduced based on principal component analysis using the "RunPCA" function with default settings. Cells were projected into 2D space using UMAP implemented in "RunUMAP" function (Parameters: dims = 1:50).

**GO enrichment analysis.** GO term enrichment was performed using clusterProfiler (v4.5.0.992)[92] with default parameters.

**Pearson correlation analysis.** For Pearson correlation analysis between *mouse* and *human* midbrain data, Homologous-HVGs sets for *mouse* and *human* were calculated based on single cell matrices from La Manno et al. using HomologySeeker. After Homologous-HVG identification, median matrices constructed by La Manno et al. (genes as rows and cell types as columns with the median value of that cell type as the matrix value) were used for Pearson correlation analysis. median matrices (x) were normalized by "log(1 + x)-rowMeans(log(1 + x))" ahead according to La Manno et al. ("rowMeans" equal to the mean value of each row). Overlapped Homologous-HVGs between *mouse* and *human* were used to calculate Pearson correlation using the "cor" function implemented in the R base package (v4.0.3).

For Pearson correlation analysis between *mouse* and *human* EHT ensembles, Homologous-HVGs sets for *mouse* and *human* were selected using HomologySeeker (based on the residual variance of each gene returned by Seurat integration using the "SCT" method (according to "EHT ensemble construction" section)). Median matrices and Pearson correlations were calculated based on corrected single-cell matrices.

**Single-cell projection.** Single-cell projection analysis was performed following the instruction of "Mapping and annotating query datasets" (https://satijalab.org/seurat/articles/integration_mapping.html).

For intra-species projection, the query single-cell dataset was normalized using the "SCTransform" function. Transfer anchors between query and reference datasets were identified using "FindTransferAnchors" function. Anchors were then

used to project the query dataset into reference using the "MapQuery" function. For inter-species projection, the PCA space of EHT ensembles was re-calculated using shared Homologous-HVGs between *mouse* and *human* EHT ensembles using the "RunPCA" function. Then anchors between query and reference were identified using the "FindTransferAnchors" function. Anchors were then used to project the query dataset into the reference using "MapQuery" function. Prediction scores were visualized in heatmap using ComplexHeatmap[93] (v2.6.2).

**Developmental trajectory inference**. Developmental trajectory of EHT ensembles was inferred using Monocle3 (v1.2.7) according to "Calculating Trajectories with Monocle 3 and Seurat" (http://htmlpreview.github.io/? https://github.com/satijalab/seurat-wrappers/blob/master/docs/monocle3.html). Briefly, *mouse* Wnt EC and *human* C1–3 (venous EC sub-clusters) were excluded from further analysis. The Seurat object was first converted to Monocle3 cell_data_set object. Unsupervised clustering of cells was performed using "cluster_cells" function (Parameter: reduction_method = "UMAP", cluster_method = "louvain"). The principal graph was learned from UMAP space using "learn_graph" function (Parameter: close_loop=F). Cell order according to pseudo time was inferred using the "order_cells" function.

For the TF expression patterns along the developmental trajectory, we fitted a local regression to the expression level for each cell at their value of pseudo time using ggplot2 (v3.3.6) ("geom_smooth" function with method = "loess") (https://ggplot2.tidyverse.org).

The developmental trajectory of single-cell RNA-seq data from Hou et al. was inferred using Monocle (v2.9.0)[16]. Briefly, the normalization factors and variability of scRNA-seq data were calculated using "estimateSizeFactors" and "estimateDispersions" functions, respectively. Only genes that expressed at least 10 cells were retained. Then the highly variable genes of scRNA-seq data calculated by "FindVariableFeatures" function from Seurat were fed into "setOrderingFilter" function to acquire features for further trajectory inference. Genes from the cell cycle GO term (GO:0007049) were filtered out from the highly variable genes to reduce the influence of the cell cycle effect. Then cells were projected into lower dimensional space using "reduceDimension" function. The final trajectory was inferred using "orderCells" function.

**TF module identification**. *Mouse* and *human* TF lists were downloaded from AnimalTFDB3.0 and HumanTFDB3.0 (http://bioinfo.life.hust.edu.cn/)[94], respectively. All TFs were selected from *mouse* and *human* Homologous-HVG lists. To identify potential TF modules, Pearson distance was calculated according to Pijuan-Sala et al.[95]. Briefly, the Pearson correlation distance between TFs was calculated as "$([1 - x]/2)^{0.5}$", where $x$ is the Pearson correlation among TFs. Then hierarchical clustering was performed using the unweighted pair group method with arithmetic mean (UPGMA), and modules were identified using the "dynamicTreeCut" function in dynamicTreeCut (v1.63-1)[54].

**Differential expression analysis**. To find DEGs, Wilcoxon Rank Sum tests implemented using "FindMarkers" function in Seurat were performed to identify DEGs. DEGs with adjusted *P* values less than 0.0001 were deemed significant. For the DEGs between *human* C6 and GJA5+ AECs, only the aggregated part (Fig. 3d, shadow in blue) of GJA5+ AECs was used for differential expression analysis. For the marker genes modules of HSC/MPP and LMPP from Huo et al.[51], DEGs between HSC/MPP or LMPP and all other cells are calculated. The top 20 upregulated DEGs that ranked by fold change were used for subsequent module score calculation.

**Module score calculation**. Module scores of TFs, marker genes, and gene sets from GO terms were estimated by using the "AddModuleScore" function in Seurat. The gene sets encompassed by *EC development* (GO:0001885), *Arterial EC differentiation* (GO:0060842), and *Blood vessel EC differentiation* (GO:0060837) were collected from AmiGO 2 (http://amigo.geneontology.org/amigo).

**Identification of potential upstream regulators of DEGs**. The upstream regulators of DEGs between pre-HEs and HEs (C7 vs. C6 in *human*) were predicted by TF enrichment analysis using ChIP-X Enrichment Analysis 3 (ChEA3; https://maayanlab.cloud/chea3/). The TF local network was constructed using the top 10 returned regulators interaction mined from the ENCODE ChIP-seq project.

**SingleCellNet analysis**. SingleCellNet (v0.1.0)[47] was used to assign *human* cells with potential identities inferred from *mice* based on differentially expressed homologous genes. Briefly, cell types classifiers were built using the "scn_train" function using *mouse* cell types as a reference (Parameter: nTopGenes = 100). *Human* cells were classified using a trained classifier using the "scn_predict" function with default settings.

**RNA velocity analysis**. Velocyto (v0.17.17)[96] was used for RNA velocity analysis of the *human* EHT ensemble. To annotate spliced, unspliced, and spanning reads in the measured cells, "run_smartseq2" and "run10x" commands were used to generate loom files for *human* STRT-seq and 10× genomics drop-based single-cell data with GRCh38/hg38 reference genome. The output loom files were combined and analyzed using the "velocyto.R" package (v0.6). RNA velocity was estimated using the "RunVelocity" function with default settings. RNA velocities were visualized on the *human* EHT ensemble using the shared nearest-neighbor graph calculated in "Dimension reduction and unsupervised clustering" section using the "show.velocity.on.embedding.cor" function (Parameter: $n = 100$, which equals neighborhood size).

**Analysis of spatial transcriptomics data**. The spatial transcriptomics matrix of the CS15 *human* embryo (slide7) was downloaded from GitHub deposited by Calvanese et al., and analyzed by Seurat. Briefly, the "SCTransform" function was used to normalize and find variable genes within the spatial transcriptomics data. Dimension reduction and unsupervised clustering were then performed according to the "Dimension reduction and unsupervised clustering" section with some modifications (Parameters: dims = 1:30 in "FindNeighbors" function, resolution = 1.2 in "FindClusters" function and dims = 1:30 in "RunUMAP" function).

**Ligand-target signaling inference**. NicheNet (v1.1.0)[61] was used to infer potential ligands that share active links with target genes (DEGs between *human* C6 and C7/late AEC). Briefly, pseudo cells from spatial transcriptomics data located in the AGM region were defined as sender cells. Potential ligands expressed by sender cells were ranked based on how well they interacted with target genes (evaluated by the Pearson correlation coefficient).

The signaling paths from ligands to target genes were inferred based on the instructions for "NicheNet Results: Ligand-Targets interesting paths" introduced by Saez lab that combine NicheNet and OmnipathR (v3.5.21) (https://github.com/saezlab/NicheNet_Omnipath/blob/master/07_LigandTargetPaths.md). The resulting pathways were visualized in Cytoscape (v3.8.2)[97].

**Statistics and reproducibility**. Data obtained from multiple experiments were reported as the mean ± SEM. An unpaired *t*-test was used to compare the means from two groups, and ANOVA was used to compare the means from three or more groups. Results with a value of *P* < 0.05 were considered statistically significant. *$P < 0.05$; **$P < 0.01$; ***$P < 0.001$.

**Reporting summary**. Further information on research design is available in the Nature Portfolio Reporting Summary linked to this article.

## Data availability

The scRNA-seq data is existing data available in GEO under accession numbers: (Zhou et al.[37], GSE135202), (Baron et al.[38], GSE112642), (Zhu et al.[18], GSE137117), (Hou et al.[16], GSE139389), (Zeng et al.[19], GSE135202), (Crosse et al.[27], GSE151877), (Calvanese et al.[20], GSE162950), (La Manno et al.[24], GSE76381), (Huo et al.[51], GSE224714). Details are listed in Supplementary Data. The highly variable homologous gene sets, differential expressed gene sets between different cell types or subclusters, GO biological pathways enriched by differentially expressed gene sets, upstream regulators of the differentially expressed genes set, NicheNet singling pathways components, primer sequences for real-time polymerase chain reaction (QPCR), and cell metadata for mouse and human ensembles are available as Excel sheets in Supplementary Data. Single-cell analysis code used in this study is available upon reasonable request.

## Code availability

HomologySeeker is openly available as an R package. The code, documentation, and examples are accessible at https://github.com/YenLab/HomologySeeker. Interfaces for *mouse* and *human* ensembles are also available.

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

## Acknowledgements

This work was supported by the National Key Research and Development Program of China (2018YFA0800200), the National Natural Science Foundation of China (31872843 and 31701160), and the National Key Research and Development Program of China (2018YFA0801003).

## Author contributions

S.M., J.H., and K.Y. designed the research; S.M. developed the HomologySeeker package. S.M. and J.H. performed the analysis. S.M., K.Q., and Q.L. performed the in vitro verification assays. S.M. and K.Y. analyzed the results, made figures, and wrote the paper. K.Y., J.H., and W.Z. commented on the paper.

## Competing interests

The authors declare no competing interests.
