## [Peer Review File · Communications Biology]

Reviewers' comments:

Reviewer #1 (Remarks to the Author):

The authors conducted hPSC differentiation to HE and defined gene regulatory network during EHT. The cross-species study is insightful. HomologySeeker algorithm can be applied to other studies like brain development, despite not being discussed deeply. Here are points that might improve the MS.

1. Specify the timepoints of each human and embryos in the figure. E.g. CS12, E10.5.
2. Purification strategy of HE population is not clear. Please describe detail in figure. Used FACS? Which CD markers?
3. GO term analysis of figure 3 is less informative. Too general and less implication in hematopoietic development. Could the authors check the expression of Hanna Mikkola's key genes RUNX1 HOXA9 MLLT3 MECOM HLF SPINK2 ALDH1A1 KCNK17 IL33 (PMID: 35418685).
4. The role of CXCL12 and its receptor CXCR4 in HSC development is well documented by Hadland (PMID: 35332125). Notch receptors and VLA-4 integrin are another major mediator of HSC development according to their study. Could the authors check the expression of Notch and integrin signals in their analysis.
5. Could the authors define surface markers that distinguish HE and AEC branching point, and sort them based on the markers. It will have a significant value to the field.
6. The HomologySeeker identified only 76 out of the top 100 biological pathways that are the same between humans and mice. It will be interesting to know whether those cell types are actually non-homologous between humans and mice, or it is just the limitation of this algorithm.

Reviewer #2 (Remarks to the Author):

In this manuscript, Mo et al. perform a comparison of published mouse and human EHT (endothelial-to-hematopoietic transition) single cell transcriptomics datasets in order to find differences and similarities. The authors report a high degree of conservation between mouse and human, but also claim some differences (see below) and report a putative point of lineage bifurcation. Finally, reanalysis of spatial transcriptomics data is performed and CXCL12 identified as an extrinsic signal acting during EHT. Functional tests in hPSCs are performed as a means to validate the role of CXCL12. The human-mouse comparison reported here is performed through a novel analysis pipeline which does not require prior annotation of cell types. The results of the analysis reported in this manuscript are of interest to the developmental hematopoiesis field, though overall provide little advancement of the existing knowledge. Moreover, I find that some assumptions and conclusions are overstated and/or dubious, which are described in detail below.

Major comments:

1. Despite the authors in this manuscript find genes and cell types as highly conserved between mouse and human, they could not find evidence of type 1-type 2 (T1-2) pre-HSC in human scRNAseq datasets. This finding contrasts with most of the results presented here. However, I am not convinced that the absence of human pre-HSCs is a real biological phenomenon, and I wonder if could be a technical issue of the analysis. For example, T1 and T2 score a high Pearson coefficient when compared to human HC/HSPC, populations which share a high degree of similarity in the transcriptome. See also Figure 1G: T1-T2 mouse pre-HSCs have a transcriptome very similar to IAC and FL HSCs. Unsupervised clustering in Supp Fig 5 identifies mouse T1 and T2 most similar to human C8 and C9 populations, with C9 scoring very high (0.62) in the T2 comparison. However, C8 and C9

also score high in the IAC and FL-HSC comparisons. The authors fail to comment on this point in the text. IAC-T1-T2-FL HSCs signatures only differ in a few genes; how is it possible then that human C9 and C10 – apparently expressing genes very similar to T2 (Figure 1G) score high in IAC and FL-HSCs but close to zero in T1-T2 comparisons (Figure 1F)? Pseudotime analysis presented in Figure 2D also suggests C8 and C9 as intermediate populations, which could likely represent pre-HSC.

Based on these evidences, I find that the claims that “no human cells equivalent to mouse pre-HSCs are identified” (line 28), “we did not observe human cell populations equivalent to mouse T1 or T2 pre-HSCs” (line 324) should be revisited in the absence of functional tests to assess for the presence of pre-HSCs in human embryos. Could this failure to find any T1/T2 similarity be potentially caused by a technical issue, possibly due to the rarity of T1 and T2 populations in these datasets, making the comparison not reach statistical significance?

2. Lines 325-327: “We observe that pre-HEs have the potential to differentiate into either HEs or IAECs, a phenomenon conserved between mouse and human”. Where are the functional data to support this claim? The authors seem to interpret “bulges” in UMAP plots as differentiation trajectories, and use this to argue that the pre-HE represents a bifurcation point during EHT. I think this is conceptually wrong as these “bulges” only represents cells slightly diverging in their transcriptome and caution should be used when using UMAPs only to make claims about cell fate choices. This is especially in light of previous data showing that the hematopoietic commitment is initiated early in development, and EHT is a unidirectional process (Swiers et al. Nat Comm 2013). Moreover, human hemogenic endothelium was shown to represent a lineage distinct from arterial endothelium (Ditadi et al Nat Cell Biol 2015).

3. Related to previous point, the authors make a number of wrong assumptions about EHT, e.g. the first sentence of the abstract: “Hemogenic endothelium (HE) arises from specialized arterial endothelial cells (AECs) during the endothelial-to-hematopoietic transition (EHT)” or line 186: “During mouse EHT, subsets of AECs choose HE or mature arterial fate (late AEC)”
What is the functional evidence that HE is a bipotent progenitor able to generate arterial endothelial cells? (See above, Swiers et al., 2013; Ditadi et al. 2015). Moreover, there is ample evidence that HE is also present within venous endothelium, which, in particular, generates erythro-myeloid progenitors (EMPs) (Frame et al Stem Cells 2016), therefore EHT is not necessarily tied to the arterial identity of their precursors.

Minor comments:

1. “These divergent trajectories towards C9 or C10 resemble recent findings in mouse, suggesting that HSCs and hematopoietic progenitors might be generated independently from the heterogeneous pre-HSPC population” (lines 166-168). What is the evidence supporting this claim? Have the authors checked any specific genes/markers? Based on the data presented here, the transcript profile of C9 and C10 appears remarkably similar.

2. Reference number 23 is erroneously reported throughout the text as “Manno et al.” It should be referred as “La Manno et al.”

3. Reference 74 is a conference abstract. Caution should be used when citing non-peer reviewed sources.

Reviewer #3 (Remarks to the Author):

The EHT process of definitive hematopoietic wave in mice has been more extensively and intensively studied than in humans, and the key cell populations as well as the molecular features altered in this

process are relatively well understood. However, the extent to which the human embryonic definitive hematopoietic process is conserved and different from that of the mouse in terms of key cell populations and molecular features remains to be urgently explored. In this manuscript, Mo et al. integrated data from multiple research groups and compared the human and mouse EHT processes in parallel using homologous highly variable gene pairs to reveal similarities and differences in cell populations in the human and mouse EHT processes. In particular, the authors identified bifurcation points in the human embryo where the arterial endothelium selects for arterial maturation fate and hematopoietic specification fate, further fine-tuned the key intercellular regulatory networks that determine fate bifurcation in conjunction with spatial transcriptome data, and validated the role of CXCL12 as an important ligand molecule in promoting hemogenic and hematopoietic potential using a human pluripotent stem cell in vitro system. The findings of this manuscript have valuable implications for understanding the human EHT process, especially the role of CXCL12 in it.

Comments:

1. In mouse, EC populations were unified as Wnt_EC, AEC, EC (Supp. Fig. 2c and Methods, Line 455), which is a few confusing to me. For example, 1) VE, Wntlo/hi VE and Wntlo/hi AE were identified in Fig. 2A of Zhu et al. 2020, but not Wnt_EC and EC shown in Fig. S2c of this study. What is their direct correspondence. It is not clear what the Wnt_EC refers to. 2) Besides, in terms of nomenclature of endothelial subgroups, arterial/artery EC (AEC) and venous EC (VEC) are reasonable, which can correspond to their molecular and/or in situ assertions. In the presence of Wnt_EC and AEC, an additional EC population are confusing. Where is the VEC population, is it EC population?
2. It is challenging to integrate datasets from different research groups, especially from different sequencing methods (e.g., full-length Smart-seq2, STRT-seq and 10x Genomics) at the same time. Thus, it is worthwhile to provide the metadata of integrated datasets, such as cell source, cluster, stage, UMAP dimensions and so on, in the form of an attached table for further analysis and visualization by broader audiences.
3. The cell metadata, including source, stage, and original clusters, is recommended to be annotated next to the heatmap in Fig. 1c. It will be more informative. For example, we can trace a subset of HSPC/HC that is more like FL-HSC but not any other cell types.
4. Why are the numbers of HVGs shown in Fig. S2b and Fig. S2c different (e.g., 3990 vs. 3900 in mouse)?
5. Which color is C5 in Fig. 3f, is it Grey? If so, it is recommended to use the identical color for the same cluster (e.g., C5 or any other cluster) in the whole manuscript. C5 population has certain hemogenic characteristic shown in Fig. 1f, 1g. Does C5 correspond to early HEC in Fig. 4f of Zeng et al. 2019?
6. Line 246-247: How to understand "the opposite expression patterns"? The data shown does not directly support this speculation, is there a better way to show? Please provide more detail explanation for it.

**Cross-species Transcriptomics Reveals Bifurcation Point During the Arterial-to-**
**Hemogenic Transition**

Shaokang Mo^{1,2,†}, Kengyuan Qu^{2,†}, Junfeng Huang^{2,†,*}, Qiwei Li², Wenqing Zhang^{1,*}, Kuangyu Yen^{2,*}

¹ Division of Cell, Developmental and Integrative Biology, School of Medicine, South China University of
Technology, Guangzhou, China

² State Key Laboratory of Experimental Hematology, National Clinical Research Center for Blood Diseases, Haihe
Laboratory of Cell Ecosystem, Institute of Hematology & Blood Diseases Hospital, Chinese Academy of Medical
Sciences & Peking Union Medical College, Tianjin, China

[†] These authors contributed equally

* To whom correspondence should be addressed:

Kuangyu Yen, PhD

E-mail: kuangyuyen@ihcams.ac.cn

Wenqing Zhang, PhD

E-mail: mczhangwq@scut.edu.cn

Abstract

[revised manuscript text omitted]

397 regulatory network among top 10 upstream regulators of up-regulated DEGs between *mouse* pre-HE and HE.
398 Network edges represent co-regulatory relationships with edges involving MYC highlighted in red.

**Figure 4. Identification of spatial ligands that facilitate cell fate choices of human pre-HEs**

**a)** Schematic of signaling pathways from spatial ligands to target genes (DEGs). **b)** Signaling pathway mediated
by CXCL12. The signal travels from CXCL12 (ligand, orange) to receptors (pink), through signaling mediators (light
blue) and regulators (blue), ending at target genes (purple) (DEGs between C6 and C7 regulated by CXCL12). **c)**
Schematic of *human in vitro* hematopoietic differentiation system from (Shen et al., 2021). **d)** Representative
flow cytometric analysis of the *in vitro*-defined HEs (CD34⁺CD144⁺CD184⁻CD73⁻) from day 4 differentiation. **e)**
Cell number of *in vitro*-defined HEs (CD34⁺CD144⁺CD184⁻CD73⁻) sorted from day 4 (left panel) differentiation
with or without CXCL12 treatment (n=3, *P<.05). **f)** Expression of *PECAM1*, *CDH5*, *TEK*, *GATA2* and *RUNX1* in cells
from day 4 with or without CXCL12 treatment. The expression level was normalized to that of β-actin. ns, not
significant. (n=3, *P<.05; **P<.01). **g)** Representative flow cytometric analysis of the *in vitro*-defined HPCs
(CD34⁺CD43⁺) from day 7 differentiation. **h)** Cell number of *in vitro*-defined HPCs (CD34⁺CD43⁺) sorted from day
7 differentiation with or without CXCL12 treatment (n=3, *P<.05). **i)** Colony-forming unit (CFU) assay of HPCs
generated with or without CXCL12 treatment. CFUs per 4x10³ cells plated (n=3, ***P<.001).

Figure 5. Cell fate choices during the pre-HE stage

During *mouse* and *human* EHT, a specific subset of AECs differentiates into pre-HEs. In response to interactions between specific cell fate determinants (like CXCL12 and PCDH7) and their corresponding receptors (like CXCR4), pre-HEs may advance toward either a hemogenic or arterial fate. Those hemogenic precursors can further develop into a hematopoietic population with multilineage differentiation capability aggregated in clusters. AEC, arterial endothelial cell; pre-HE, pre-hemogenic endothelium; HE, hemogenic endothelium; IAC, intra-aortic cluster.

REVIEWERS' COMMENTS:

Reviewer #1 (Remarks to the Author):

The authors adequately addressed the points.

Reviewer #2 (Remarks to the Author):

The authors have satisfactorily replied to all of my concerns. Substantial effort has been put into adding new analysis in the revised version of this manuscript. Therefore, I support publication of the revised manuscript.

Reviewer #3 (Remarks to the Author):

The authors have addressed all queries that I raised. I think the revised manuscript is clearer and more convincing.